# Concordance of Non-Alcoholic Fatty Liver Disease and Associated Factors among Older Married Couples in China

**DOI:** 10.3390/ijerph20021426

**Published:** 2023-01-12

**Authors:** Xueli Yuan, Wei Liu, Wenqing Ni, Yuanying Sun, Hongmin Zhang, Yan Zhang, Peng Yin, Jian Xu

**Affiliations:** 1Department of Elderly Health Management, Shenzhen Center for Chronic Disease Control, Shenzhen 518020, China; 2National Center for Chronic and Noncommunicable Disease Control and Prevention, Chinese Center for Disease Control and Prevention, Beijing 100050, China

**Keywords:** older couples, non-alcoholic fatty liver disease, spousal concordance, risk factors, lifestyle

## Abstract

Background: Non-alcoholic fatty liver disease (NAFLD) is one of the most common liver diseases which affects mainly middle-aged and older adults, resulting in a considerable disease burden. Evidence of concordance on NAFLD and lifestyle factors within older married couples in China is limited. This study aimed to evaluate spousal concordance regarding lifestyle factors and NAFLD among older Chinese couples. Methods: We conducted a cross-sectional study using data from 58,122 married couples aged 65 years and over recruited from Shenzhen, China during 2018–2020. Logistic regression analyses were used to estimate the reciprocal associations in NAFLD within couples after incremental adjustment for potential confounders. Results: There was a marked concordance regarding NAFLD among older married couples in our study. After adjustment for confounders, the odds of having NAFLD were significantly related to the person’s spouse also having NAFLD (1.84 times higher in husbands and 1.79 times higher in wives). The spousal concordance of NAFLD was similar, irrespective of gender. Couples with both a higher educational level and abdominal obesity were more likely to have a concordance of NAFLD compared to couples with both a lower educational level and no abdominal obesity, respectively (*p* < 0.05). Conclusion: Our results indicated that health care professionals should bear in mind the marked spousal concordance with respect to risk factors and NAFLD for the prevention and early detection of the highly prevalent disease in older Chinese adults.

## 1. Introduction

Non-alcoholic fatty liver disease (NAFLD) is one of the most common liver diseases which affects mainly middle-aged and older adults, resulting in a considerable disease burden with variable socio-economic implications [1]. The prevalence of NAFLD in the general Chinese population is 18.5–23.1%, which was significantly higher than the global average (14.3–17.8%) in 2019 [2]. It is estimated that more than 120 million Chinese people aged 55 years and over were affected by NAFLD in 2019, accounting for more than 33% of the total disease of this population [2]. There were 190 million people aged 65 and above in China in 2020 [3], and this figure is expected to increase to 366 million by 2050 [4,5]. The large number of older Chinese adults with NAFLD will sequentially create a significant health burden in China in the future. It has been indicated that behavioral factors, including non-smoking, high-fiber food intake, and moderate physical activity, may contribute substantially to preventing and treating NAFLD [1,6,7].

It is well-established that couples can profoundly affect each other’s physical and mental health [8,9] as they usually live together and influence one another’s behavioral habits [10,11]. Accumulating epidemiological studies have shown a spousal concordance of lifestyles and health conditions, with a focus primarily on overweight/obese individuals, diabetes, hypertension, hyperlipidemia, and cardiovascular diseases [8,10,11,12,13,14,15]. However, data on common liver diseases such as NAFLD among couples were limited. Particularly in China, where NAFLD is highly prevalent, whether spouses share the similarity and what the potential factors associated with the similarity are remain unclear. Answers to these questions can provide evidence for couples-based intervention measures aiming to detect and prevent the progress of this highly prevalent disease in older Chinese adults. 

Therefore, we carried out this study to investigate the spousal concordance of NAFLD among married couples aged 65 years and over to explore gender differences in spousal associations and to identify the potential factors associated with this concordance using data from the Shenzhen Healthy Ageing Research (SHARE) study in Shenzhen, China.

## 2. Methods

### 2.1. Study Population

Our study population was obtained from the Shenzhen Healthy Ageing Research (SHARE) study, which examined approximately 63.9% of the Shenzhen older adult population aged 65 and over for the period 2018–2020. The full details of the survey procedures and baseline variables have been described in previous publications [16]. We used the following criteria for couples in our study: (1) both spouses were enrolled in the survey, and (2) complete information on the sociodemographic and lifestyle variables, as well as abdominal B-ultrasound examination results, was provided for both spouses. A total of 59,364 pairs of spouses aged 65 years and above were enrolled in the SHARE study during 2018–2020. We excluded 1047 pairs of spouses with incomplete answers and 195 pairs of spouses with outliers for the key variables (body mass index, BMI, of <15 kg/m^2^ or BMI of >40 kg/m^2^ and waist circumference, WC, of <40 cm or WC of >200 cm). A total of 58,122 pairs of spouses (116,244 participants) were included in the final analysis.

The study was approved by the Ethics Committee of Shenzhen Center for Chronic Disease Control.

### 2.2. Measurements

Baseline data on the participants’ sociodemographic and lifestyle variables were collected and categorized by age group (65–69, 70–74, 75–79, and 80 or older), educational level (low: ≤9 years, middle: 9–12 years, and high: >12 years), smoking status (never-smoker and former or current smoker), physical activity (yes or no), abdominal obesity by waist circumference (yes (male: waist measurement of >90 cm, female: waist measurement of >80 cm) or no), BMI (<18 kg/m^2^, 18–23.9 kg/m^2^, 24–27.9 kg/m^2^, and ≥28 kg/m^2^), and self-rated health (good, fair, or poor). 

The primary exposure and outcome were whether the couple had NAFLD. The participants’ abdomens were examined by color Doppler ultrasound. The examiner had the participants assume a supine position and fully exposed their abdomens, focusing on the shape and size of their livers and gallbladders and paying attention to the nature of the echo and its relationship with the surrounding tissues, while also recording the location and number of lesions and the shape and dilatation of hepatic bile tubules. Diagnoses were made according to the guidelines of prevention and treatment for NAFLD [17,18]. Ultrasonography is the primary imaging tool to detect NAFLD in China [19]. We defined having NAFLD as those participants who did not drink excessive alcohol (male: daily alcohol consumption of <30 mg and female: daily alcohol consumption of <20 mg) in the past year and whose hepatitis B ultrasound imaging findings met the diagnostic criteria of diffuse fatty liver for which there was no other explanation [17,18].

### 2.3. Statistical Analysis

The participants’ baseline characteristics are shown as numbers (percentages) for the categorical variables. We used the χ^2^ test of independent groups to test the differences in NAFLD across the various characteristic groups and the McNemarχ^2^ test to explore the differences within couples in lifestyle factors and NAFLD. 

We performed logistic regression analyses to estimate the reciprocal associations in NAFLD within couples using the husbands/wives having NAFLD as the outcome variable and the wives/husbands having NAFLD as the principal exposure variable. To illustrate potential confounding factors, we fit the following four models in the analysis: model 1 was crude, without any adjustment; age and educational level were adjusted in model 2; smoking and physical activity were further added in model 3; and abdominal obesity and self-rated health were added in model 4. Odds ratios (ORs) with 95% confidence intervals (CIs) were computed for these models.

Stratified analyses by gender were carried out for the total sample and for different age groups. Interaction tests were applied to assess gender differences. In order to explore the associated factors of NAFLD concordance between husbands and wives, we performed logistic regression analyses using concordant couples as a reference group and we estimated the ORs for discordant couples and couples who were not diagnosed as having NAFLD and who fit the social demographic characteristics and behavior and lifestyle factors. In addition, we conducted a sensitivity analysis among couples with different places of origin (different provinces, cities, and districts/counties) to rule out the potential effect of childhood lifestyles on adulthood chronic diseases [20,21].

All statistical analyses were conducted using SAS software (version 9.4, SAS Institute Inc., Cary, NC, USA) and all graphs were plotted with R (version 4.1). All tests were two-sided, and a *p* value of <0.05 was considered statistically significant.

## 3. Results

### 3.1. Characteristics of the Study Population

The number of participants classified with NAFLD was 14,622 (25.2%), or 20,082 (34.6%) pairs of husbands and wives, respectively. Th characteristics of the overall study participants and those with NAFLD are shown in Table 1. Both the husbands and wives with NAFLD were younger (65–69 years of age: 25.8% men and 34.9% women; *p* < 0.001; and 70–74 years of age: 25.8% men and 36.2% women; *p* < 0.001), more likely to have abdominal obesity (41.3% men and 49.4% women; *p* < 0.001), and more likely to have middle or high educational levels (middle: 27.8% men vs. 36.1% women; *p* < 0.001; and high: 30.3% men vs. 35.2% women; *p* < 0.001) than those without NAFLD. As shown in Appendix A, the main comorbidities of NAFLD were hypertension (63.0%), diabetes (30.9%), and hyperlipidemia (55.6%).

### 3.2. Spousal Concordance for NAFLD and Associated Factors

As shown in Table 2, the prevalence values of current smoking, physical inactivity, overweight/obese, and abdominal obesity for both couples were 0.2%, 6.2%, 25.3%, and 21.8%, respectively. Husbands were less likely to be physical inactive (OR_MP_ (matched pairs odds ratio) = 0.74, 95% CI: 0.71–0.77) and to have abdominal obesity (OR_MP_ = 0.67, 95% CI: 0.65–0.68) compared to their wives. Husbands had significantly higher odds of smoking (OR_MP_ = 35.69, 95% CI: 33.20–38.36) compared to their wives. The prevalence of NAFLD among both partners in a couple was 11.3%. Husbands had significantly lower odds of having NAFLD (OR_MP_ = 0.60, 95% CI: 0.58–0.61) compared to their wives.

Significant concordance was observed within couple pairs for NAFLD (OR_adjusted_ = 1.81; 95% CI, 1.76–1.86) in the crude and incrementally adjusted models (Table 3). Husbands whose wives had NAFLD showed an increased risk of having NAFLD (OR_adjusted_ = 1.84, 95% CI: 1.77–1.92), and this risk was statistically significant for the wives (OR_adjusted_ = 1.79, 95% CI: 1.71–1.86). The results of the interaction tests indicated that the spousal concordance for NAFLD was similar, irrespective of gender (*p* > 0.05 for the interaction). 

Table 4 shows the results of the multivariate logistic regression for the associations between concordant sociodemographic and lifestyle variables and concordant incidence of NAFLD among spouses (n = 28,104 pairs). Couples with both higher educational levels, abdominal obesity, and poor self-rated health were 1.30, 3.10, and 1.30 times more likely to have concordant NAFLD compared to couples with both lower educational levels, no abdominal obesity, and good self-rated health, respectively (*p* < 0.05).

### 3.3. Stratification Analysis by Age

We further investigated the spousal associations of NAFLD in four age groups (Figure 1). Among all the age groups, the husband’s NAFLD was significantly associated with the wife’s NAFLD. The extent of the association with NAFLD from husbands to wives appeared to be similar, as did the reverse (aged 65–69: OR_adjusted_ = 1.73 (95% CI, 1.62–1.86) vs. 1.79 (1.67–1.92); *p* = 0.50 for the interaction; aged 70–74: 1.77 (1.58–1.98) vs. 1.85 (1.66–2.08); *p* = 0.57 for the interaction; aged 75–79: 2.38 (1.87–3.02) vs. 2.60 (2.03–3.33); *p* = 0.61 for the interaction; and aged 80 and above: 2.20 (1.76–2.75) vs. 2.27 (1.83–2.83); *p* = 0.84 for the interaction), indicating no gender specificity of spousal health concordance for all four age groups. 

### 3.4. Sensitivity Analysis

To rule out the potential effect of childhood lifestyle on older adults with NAFLD, we analyzed the spousal concordance of NAFLD among older couples with different places of origin as assessed by those born in different provinces, cities, or districts/counties (Figure 2). We found that even if the couples were originally from different provinces (n = 1772 pairs), cities (n = 5071 pairs), or districts/counties (n = 7414 pairs), we found that the results remained consistently significant, with no substantial change. The gender specificity of spousal health concordance showed no statistical significance. 

## 4. Discussion

Based on the studied 58,122 pairs of older Chinese couples, we found spousal concordance for NAFLD, irrespective of gender. Higher educational levels, abdominal obesity, and poor self-rated health were associated with the spousal concordance of NAFLD.

Although spousal health concordance has been reported in many studies, previous studies have primarily focused on cardiovascular diseases, metabolic syndrome, and health-related behaviors [8,11,12,13,14,22]. To our knowledge, this study is the first analysis indicating spousal concordance of NAFLD, stratified by gender and age and adjusted for potential covariates. The findings of spousal concordance for NAFLD may be explained by the shared resource hypothesis [23,24,25], the theory of emotional contagion [15], and the caregiver burden hypothesis [10,26], which were shared in previous studies examining the concordance of other chronic conditions. A large number of epidemiological studies [27,28,29,30] have shown that NAFLD is associated with insufficient physical exercise, a high BMI, and abdominal obesity. Consistent with previous studies [8,12,31], our study found that there was a high level of concordance regarding behavioral and lifestyle factors among older married couples, which could lead to an increase in the prevalence of NAFLD among this population. 

NAFLD is the most common liver disease in China [32,33], and the burden of NAFLD-related advanced liver disease is expected to increase substantially [32]. In line with previous studies, we found a high concordance of lifestyle factors and NAFLD between older married couples in China [8,11,12,13,14]. Our findings shed light on why it is important for medical professionals to consider applying couples-based interventions to individuals with NAFLD. Common health promotion activities may need to include the involvement of one’s spouse and consider the couple together. Moreover, recognizing that patients and their spouses have common disease risks may encourage them to participate in a physical examination together and improve their lifestyle habits, such as implementing a balanced diet and exercise.

Previous studies on the gender variation in spousal health concordance have been inconsistent [22,34,35]. Some research [22,36,37] has indicated that husbands were more susceptible to spousal chronic diseases than wives, while others have indicated the contrary [13,31,38]. However, our study does not provide evidence to support the gender variation in spousal concordance for NAFLD. In our study, both husbands and wives showed significant health consistency with their partners. We thus speculate that the health status of a husband may be similar to that of his wife because, in this elderly Chinese population, a husband often depends on the care of his spouse. If a wife is ill, her husband may not receive proper care, which can have a negative effect on her husband’s health [22,36]. In addition, it is also possible that wives are more vulnerable to the health of their husbands. This may be associated with the fact that women are typically more sensitive to the negative emotions of others in the face of disease pressure, and they often assume the responsibility of taking care of their partners, which, in turn, may worsen their own health [39,40]. In general, many senior couples take nonsteroidal anti-inflammatory drugs (NSAIDs) for a variety of comorbidities that may lead to drug-induced liver injury (DILI) [41]. DILI, NSAIDs, and the idea of pairwise factor analysis may also be one of the potential factors causing the spousal concordance of NAFLD among older adults. The differences in gender roles of the different studies may result from the mixed results of cultural differences and other subtle background factors [42]. Future research is necessary to comprehensively address the impact of gender on different spouses.

We found that compared with couples with low education, couples in which at least one of spouses with middle or high education had a higher risk of NAFLD. This may be explained by older adults with a higher education level having higher social capital and socioeconomic status, which may be an independent risk factor for NAFLD [43,44]. It is well-known that obesity is one of the most important risk factors for NAFLD, especially abdominal obesity [1,32]. Our study also found that abdominal obesity (in both partners or in one of them) is a risk factor for NAFLD for both the husband and wife. Some studies have shown that unhealthy lifestyles in childhood increase the risk of chronic disease in adulthood [20,21]. Being overweight in childhood and adolescence is associated with an increased risk for NAFLD later in life [28]. A systematic review indicated that regional economics and environment were important factors in NAFLD progression [32]. Therefore, we conducted sensitivity analyses among older couples with different places of origin to rule out the potential effect of childhood lifestyle on older adults with NAFLD. The results were consistent with the main results.

Apart from the large sample size of this study, another advantage is that the diagnosis of the study outcome was relatively reliable, largely based on the results of the B ultrasounds during the physical examinations, while the outcomes of other studies are self-reported by participants and have great recall bias. Several limitations must also be considered. First, the cross-sectional nature of the study design resulted in causal relationships being unable to be inferred. Second, we were unable to assess some factors which may have affected spousal health, such as spousal intimacy, whether spouses were primary caregivers for each other, and genetic factors, due to data availability. Finally, the specific study area limited the generalizability of our results, though it included a considerably large sample size. Further studies are needed to confirm our results in other populations.

## 5. Conclusions

There was a marked concordance regarding NAFLD among older married couples in our study, and such spousal relationships showed no gender specificity. Our results indicated that healthcare professionals should bear in mind the marked spousal concordance with respect to risk factors and NAFLD for the prevention and early detection of this highly prevalent disease in older Chinese adults.

## Figures and Tables

**Figure 1 ijerph-20-01426-f001:**
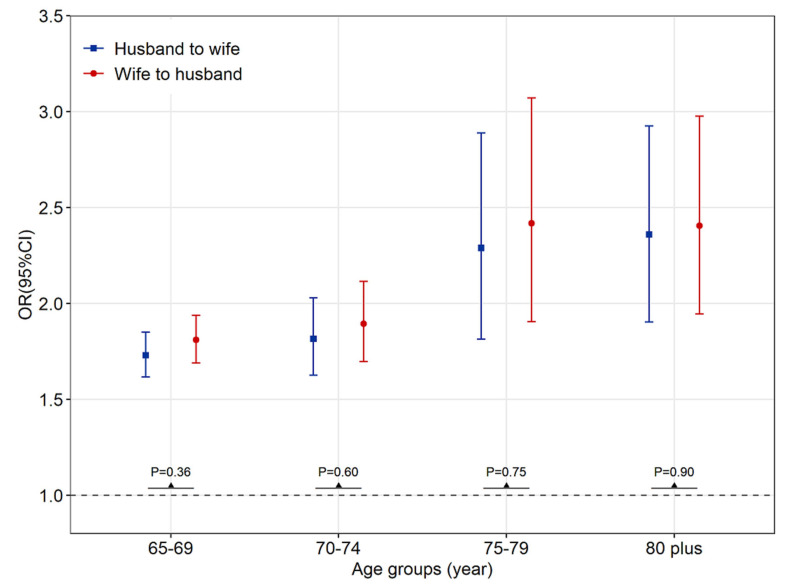
Reciprocal NAFLD association by gender among different age groups for the period 2018–2020. Note: *p*-value for gender interaction. Abbreviations: OR: odds ratio; CI: confidence intervals.

**Figure 2 ijerph-20-01426-f002:**
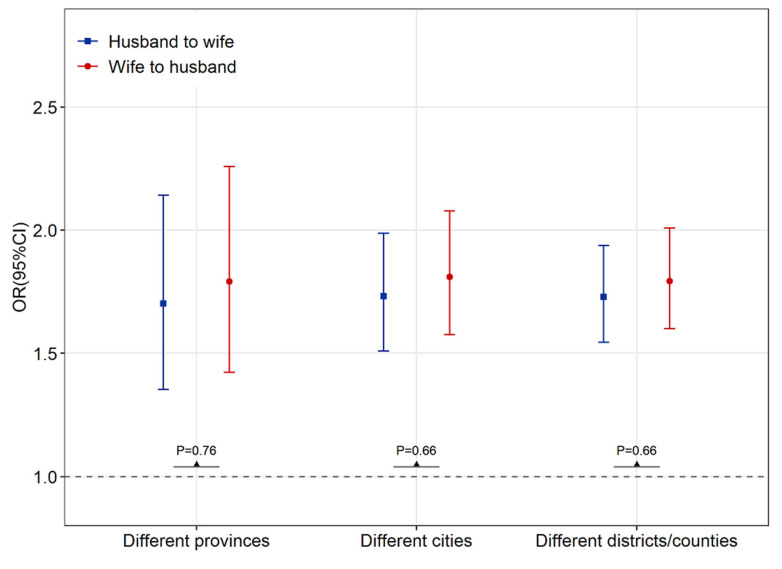
Reciprocal NAFLD association among older couples with different places of origin, stratified by those born in different provinces, cities, or districts/counties, for the period 2018–2020. Note: *p*-value for gender interaction. Abbreviations: OR: odds ratio; CI: confidence intervals.

**Table 1 ijerph-20-01426-t001:** Characteristics of the study participants categorized by gender (husband and wife).

Characteristics	Husband				Wife			
	Overall		NAFLD		Overall		NAFLD	
	N	%	N	%	N	%	N	%
Total	58,122	100	14,622	12.6	58,122	100	20,082	17.3
Age group (year)								
65–69	22,271	38.3	5748	25.8	34,462	59.3	12,034	34.9
70–74	19,520	33.6	5039	25.8	14,734	25.4	5335	36.2
75–79	9587	16.5	2326	24.3	5928	10.2	1950	32.9
80+	6744	11.6	1509	22.4	2998	5.2	763	25.5
*p*-value	NA		**<0.001**		NA		**<0.001**	
Educational level								
Low	34,041	58.6	7696	22.6	40,527	69.7	13,782	34.0
Middle	14,713	25.3	4087	27.8	12,186	21.0	4395	36.1
High	9368	16.1	2839	30.3	5409	9.3	1905	35.2
*p*-value	NA		**<0.001**		NA		**<0.001**	
Smoking status								
Never	38,707	66.6	10,089	26.1	57,602	99.1	19,906	34.6
Former	9049	15.6	2330	25.7	110	0.2	37	33.6
Current	10,366	17.8	2203	21.3	410	0.7	139	33.9
*p*-value	NA		**<0.001**		NA		0.943	
Physical activity								
Yes	8137	14.0	1844	22.7	9732	16.7	3413	35.1
No	49,985	86.0	12,778	25.6	48,390	83.3	16,669	34.4
*p*-value	NA		**<0.001**		NA		0.239	
BMI (kg/m^2^)								
<18.5	1915	3.3	31	1.6	1878	3.2	42	2.2
18.5–23.9	28,340	48.8	3734	13.2	28,432	48.9	5819	20.5
24–27.9	22,924	39.4	7978	34.8	21,357	36.7	9871	46.2
>28	4943	8.5	2879	58.2	6455	11.1	4350	67.4
*p* value	NA		**<0.001**		NA		**<0.001**	
Abdominal obesity								
No	35,331	60.8	5205	14.7	30,268	52.1	6328	20.9
Yes	22,791	39.2	9417	41.3	27,854	47.9	13,754	49.4
*p*-value	NA		**<0.001**		NA		**<0.001**	
Self-rated health								
Good	55,210	95.0	13,932	25.2	54,677	94.1	18,946	34.7
Fair	918	1.6	224	24.4	978	1.7	312	31.9
Poor	1994	3.4	466	23.4	2467	4.2	824	33.4
*p*-value	NA		0.147		NA		0.095	

Abbreviations: NAFLD: non-alcoholic fatty liver disease; NA: not applicable. The bold: *p* < 0.05.

**Table 2 ijerph-20-01426-t002:** Spousal concordance for lifestyle factors and NAFLD among the 58,122 married couples.

Characteristic	Both		Husbands Only		Wives Only		Neither		OR_MP_ (95% CI)	*p*-Value
	N	%	N	%	N	%	N	%		
Risk factors										
Current smoking	123	0.2	10,243	17.6	287	0.5	47,469	81.7	35.69 (33.20,38.36)	**<0.001**
No physical activity	3628	6.2	4509	7.8	6104	10.5	43,881	75.5	0.74 (0.71,0.77)	**<0.001**
Overweight/obese	14,692	25.3	13,175	22.7	13,120	22.6	17,135	29.5	1.00 (0.98,1.03)	0.735
Abdominal obesity	12,684	21.8	10,107	17.4	15,170	26.1	20,161	34.7	0.67 (0.65,0.68)	**<0.001**
Diseases										
NAFLD	6600	11.4	8022	13.8	13,482	23.2	30,018	51.6	0.60 (0.58,0.61)	**<0.001**

Abbreviations: NAFLD: non-alcoholic fatty liver disease; OR_MP_: matched pairs odds ratio. The bold: *p* < 0.05.

**Table 3 ijerph-20-01426-t003:** Reciprocal NAFLD association among the 58,122 older couples for the period 2018–2020.

Outcomes	Model Adjusting for Gender, Total		Husband to Wife		Wife to Husband		*p* Value for Gender Interaction
	OR (95% CI)	*p*-Value	OR (95% CI)	*p*-Value	OR (95% CI)	*p*-Value	
NAFLD							
Model 1	1.83 (1.78–1.88)	**<0.001**	1.83 (1.76–1.90)	**<0.001**	1.83 (1.76–1.90)	**<0.001**	1.000
Model 2	1.81 (1.77–1.86)	**<0.001**	1.82 (1.75–1.89)	**<0.001**	1.82 (1.76–1.90)	**<0.001**	0.912
Model 3	1.82 (1.77–1.87)	**<0.001**	1.83 (1.76–1.90)	**<0.001**	1.82 (1.76–1.90)	**<0.001**	0.896
Model 4	1.81 (1.76–1.86)	**<0.001**	1.79 (1.71–1.86)	**<0.001**	1.84 (1.77–1.92)	**<0.001**	0.287

Notes: model 1 was unadjusted; model 2 was adjusted for age and educational level; model 3 additionally adjusted for behavioral covariates, including physical activity and smoking; and model 4 additionally adjusted for abdominal obesity and self-rated health. Abbreviations: NAFLD: non-alcoholic fatty liver disease; OR: odds ratio; CI: confidence intervals. The bold: *p* < 0.05.

**Table 4 ijerph-20-01426-t004:** Logistic regression for the effect of different factors on concordant NAFLD among the 28,104 older couples.

Characteristic	Overall N (%)	NAFLD, N (%)		Husband/Wife Only vs. Both	
		Husband/Wife Only	Both	OR (95% CI)	*p*-Value
Total	28,104 (100)	21,504 (100)	6600 (100)		
Age: <70 years old					
Both	10,025 (35.7)	7692 (35.8)	2333 (35.3)	ref	
Husband/wife only	7598 (27.0)	5854 (27.2)	1744 (26.4)	0.99 (0.92,1.07)	0.858
Neither	10,481 (37.3)	7958 (37.0)	2523 (38.2)	0.99 (0.93,1.06)	0.740
Educational level: high					
Both	1799 (6.4)	1296 (6.0)	503 (7.6)	ref	
Husband/wife only	4153 (14.8)	3108 (14.5)	1045 (15.8)	0.87 (0.76,0.99)	**0.030**
Neither	22,152 (78.8)	17,100 (79.5)	5062 (76.5)	0.76 (0.68,0.85)	**<0.001**
Smoking status: never					
Both	18,524 (65.9)	14,082 (65.5)	4442 (67.3)	ref	
Husband/wife only	9471 (33.7)	7340 (34.1)	2131 (32.3)	0.90 (0.85,0.96)	**<0.001**
Neither	109 (0.4)	82 (0.4)	27 (0.4)	0.97 (0.62,1.51)	0.879
Physical activity: yes					
Both	21,284 (75.7)	16,209 (75.4)	5075 (76.9)	ref	
Husband/wife only	5172 (18.4)	4037 (18.8)	1135 (17.2)	0.90 (0.84,0.97)	**0.007**
Neither	1648 (5.9)	1258 (5.9)	390 (5.9)	1.00 (0.89,1.13)	0.986
Abdominal obesity: no					
Both	6219 (22.1)	5238 (24.4)	981 (14.9)	ref	
Husband/wife only	13,327 (47.4)	10,817 (50.3)	2510 (38.0)	1.26 (1.16,1.37)	**<0.001**
Neither	8558 (30.5)	5449 (25.3)	3109 (47.1)	3.10 (2.86,3.37)	**<0.001**
Self-rated health: good					
Both	25,421 (90.5)	19,423 (90.3)	5998 (90.9)	ref	
Husband/wife only	2398 (8.5)	1883 (8.8)	515 (7.8)	0.86 (0.77,0.95)	**0.004**
Neither	285 (1.0)	198 (0.9)	87 (1.3)	1.30 (1.00,1.69)	**0.049**

Abbreviations: NAFLD: non-alcoholic fatty liver disease; OR: odds ratio; CI: confidence intervals; ref: reference. The bold: *p* < 0.05.

## Data Availability

The original contributions presented in the study are included in the article/Appendix A, and further inquiries may be directed to the corresponding authors.

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
