# Peer review of "Concordance of Non-Alcoholic Fatty Liver Disease and Associated Factors among Older Married Couples in China"

_ijerph, 2023, doi:10.3390/ijerph20021426_

Round 1

Reviewer 1 Report

In this work, the authors study non-alcoholic fatty liver disease among older couples in china.

The work is well written. I have the following comments and queries.   1. Please rewrite the background portion of the abstract. It is unclear in the present state. 2. Can you please include the selection criteria for patients? What were the comorbidities? Did you track patients with liver injury and excluded them from the study? 3.The analysis is clear to understand. 4. In your discussion. I suggest you highlight the issue of drug induced liver injury. In general, many senior couples are on NSAIDs for a variety of comorbidities that may lead to DILI. One important work you need to cite:   Datta A, Flynn NR, Barnette DA, Woeltje KF, Miller GP, Swamidass SJ (2021) Machine learning liver-injuring drug interactions with non-steroidal anti-inflammatory drugs (NSAIDs) from a retrospective electronic health record (EHR) cohort. PLoS Comput Biol 17(7): e1009053. https://doi.org/10.1371/journal.pcbi.1009053    This work uses a modified linear model to account for pairwise dependent variables for DILI. In the discussion, you need to highlight DILI, NSAIDs and the idea of pairwise factor analysis. This would significantly increase the quality of the manuscript.

Author Response

Dear Reviewer,

Thank you for reviewing our manuscript. We really appreciate your valuable comments and we have revised our manuscript accordingly. Revised sections are highlighted in red font in the manuscript. Please see our point-to-point responses as below.

In this work, the authors study non-alcoholic fatty liver disease among older couples in china. The work is well written. I have the following comments and queries.

  1. Please rewrite the background portion of the abstract. It is unclear in the present state.

Response: Thanks for your suggestion, and we have rewritten the background portion of the abstract in line 11-15 in the revision as follows: “Non-alcoholic fatty liver disease (NAFLD) is one of the most common liver diseases which affects mainly middle-aged and older adults, resulting in a considerable disease burden. Evidence of concordance on NAFLD and lifestyle factors within older married couples in China is limited. This study aimed to evaluate spousal concordance regarding lifestyle factors and NAFLD among Chinese older couples.”

  1. Can you please include the selection criteria for patients? What were the comorbidities? Did you track patients with liver injury and excluded them from the study?

Response: Thanks for your suggestion. We have described the selection criteria for patients in method section in line 82-92 in the revision as follows: “The main exposure and outcome were whether the couple had NAFLD. The abdomen was examined by color Doppler ultrasound. The examiner took supine position and fully exposed the abdomen, focusing on the shape and size of liver and gallbladder, paying attention to the nature of echo and its relationship with surrounding tissues, and recording the location and number of lesions, the shape and dilatation of hepatic bile tubules. The diagnosis was made according to guidelines of prevention and treatment for NAFLD [1,2]. Ultrasonography is the primary imaging tool to detect NAFLD in China [3]. We defined NAFLD as participants who did not drink excessive alcohol (male: daily alcohol consumption < 30mg; female: daily alcohol consumption <20mg) in the past year and hepatic B-ultrasound imaging findings met the diagnostic criteria of diffuse fatty liver and there was no other reason to explain [1,2].”

We have added Table E1 showing the distribution of the comorbidities among 34,704 NAFLD patients and added texts in line 127-129 as following: “As shown in Table E1, the main comorbidities of NAFLD were hypertension (63.0%), diabetes (30.9%) and hyperlipidemia (55.6%).”.

Table E1. The distribution of the comorbidities among 34,704 NAFLD patients.

Diseases

Overall, N (%)

Husband, N(%)

Wife, N (%)

NAFLD

34704 (100)

14622 (100)

20082 (100)

Comorbidities

Hypertension

21856 (63.0)

9404 (64.3)

12452 (62.0)

Diabetes

10734 (30.9)

4646 (31.8)

6088 (30.3)

Hyperlipidemia

19280 (55.6)

8365 (57.2)

10915 (54.4)

Any of three diseases

29795 (85.9)

12724 (87.0)

17071 (85.0)

Abbreviations: NAFLD: non-alcoholic fatty liver disease.

In this study, fatty liver was mainly diagnosed by abdominal color ultrasound without more detailed physiological and biochemical examination, so we did not track patients with liver injury and excluded them from the study. We agree with the reviewer that patients with liver injury is an important issue for NAFLD, however, the study population from the physical examination population is relatively healthy, and the incidence of liver injury is relatively low. To reflect this issue, we have added the paragraph discussing the issue of drug induced liver injury, incorporating comment No. 4, in line 225-229 as following: “In general, many senior couples are on nonsteroidal anti-inflammatory drugs (NSAIDs) for a variety of comorbidities that may lead to drug induced liver injury (DILI) [4]. DILI, NSAIDs and the idea of pairwise factor analysis may also be one of the potential factors causing spousal concordance in NAFLD among older adults.”               

[1]   Chalasani N, Younossi Z, Lavine JE, Diehl AM, Brunt EM, Cusi K, et al. The diagnosis and management of non-alcoholic fatty liver disease: practice guideline by the American Gastroenterological Association, American Association for the Study of Liver Diseases, and American College of Gastroenterology. Gastroenterology (2012) 142(7):1592-609. doi: 10.1053/j.gastro.2012.04.001.

[2]   National Workshop on Fatty Liver and Alcoholic Liver Disease, Chinese Society of Hepatology, Chinese Medical Association, Fatty Liver Expert Committee, Chinese Medical Doctor Association. Guidelines of prevention and treatment for nonalcoholic fatty liver disease: a 2018 update. Chinese Journal of Hepatology (2018) 26(3):195-203. doi: 10.3760/cma.j.issn.1007-3418.2018.03.008.

[3]   Zhou F, Zhou J, Wang W, Zhang XJ, Ji YX, Zhang P, et al. Unexpected Rapid Increase in the Burden of NAFLD in China From 2008 to 2018: A Systematic Review and Meta-Analysis. Hepatology (2019) 70(4):1119-33. doi: 10.1002/hep.30702.

[4] Datta, A.; Flynn, N.R.; Barnette, D.A.; Woeltje, K.F.; Miller, G.P.; Swamidass, S.J. Machine learning liver-injuring drug inter-actions with non-steroidal anti-inflammatory drugs (NSAIDs) from a retrospective electronic health record (EHR) cohort. Plos Comput Biol 2021, 17, e1009053, doi:10.1371/journal.pcbi.1009053.

3.The analysis is clear to understand.

Response: Thank you for the positive comments.

  1. In your discussion. I suggest you highlight the issue of drug induced liver injury. In general, many senior couples are on NSAIDs for a variety of comorbidities that may lead to DILI. One important work you need to cite: Datta A, Flynn NR, Barnette DA, Woeltje KF, Miller GP, Swamidass SJ (2021) Machine learning liver-injuring drug interactions with non-steroidal anti-inflammatory drugs (NSAIDs) from a retrospective electronic health record (EHR) cohort. PLoS Comput Biol 17(7): e1009053. https://doi.org/10.1371/journal.pcbi.1009053 This work uses a modified linear model to account for pairwise dependent variables for DILI. In the discussion, you need to highlight DILI, NSAIDs and the idea of pairwise factor analysis. This would significantly increase the quality of the manuscript.

Response: Thanks for your suggestion, and We have added the paragraph discussing the issue of drug induced liver injury in line 225-229 as following: “In general, many senior couples are on nonsteroidal anti-inflammatory drugs (NSAIDs) for a variety of comorbidities that may lead to drug induced liver injury (DILI) [1]. DILI, NSAIDs and the idea of pairwise factor analysis may also be one of the potential factors causing spousal concordance in NAFLD among older adults.”

[1] Datta, A.; Flynn, N.R.; Barnette, D.A.; Woeltje, K.F.; Miller, G.P.; Swamidass, S.J. Machine learning liver-injuring drug inter-actions with non-steroidal anti-inflammatory drugs (NSAIDs) from a retrospective electronic health record (EHR) cohort. Plos Comput Biol 2021, 17, e1009053, doi:10.1371/journal.pcbi.1009053.

Once again, we thank you for the valuable comments and suggestions and we look forward to hearing from you.

Sincerely

Peng Yin, PhD

Professor of Epidemiology

National Center for Chronic and Noncommunicable Disease Control and Prevention

Chinese Center for Disease Control and Prevention

Reviewer 2 Report

This study shows a concordance regarding NAFLD among older married couples. It is considered appropriate in terms of research method and interpretation.

Please consider making some minor modifications as follows.

-In line 67, Appropriate conjunctions or expressions must be added in front of 58,122 pairs of spouses.

-In table 4, it would be easier to interpret if the level of education was expressed as high instead of low. In this case Both and Neither must be reciprocally changed. The rest of the variables can be changed in the same way.

Author Response

Dear Reviewer,

Thank you for reviewing our manuscript. We really appreciate your valuable comments and we have revised our manuscript accordingly. Revised sections are highlighted in red font in the manuscript. Please see our point-to-point responses as below.

This study shows a concordance regarding NAFLD among older married couples. It is considered appropriate in terms of research method and interpretation. Please consider making some minor modifications as follows.

  1. In line 67, Appropriate conjunctions or expressions must be added in front of 58,122 pairs of spouses.

Response: Thank you for the comment. As suggested, we have revised the sentence to “A total of 58,122 pairs of spouses (116,244 participants) were included in the final analysis.” in line 71-72.

  1. In table 4, it would be easier to interpret if the level of education was expressed as high instead of low. In this case Both and Neither must be reciprocally changed. The rest of the variables can be changed in the same way.

Response: Thank you for these insightful comments and suggestions. We have revised the results in table 4 according to the reviewer’s suggestion.

Once again, we thank you for the valuable comments and suggestions and we look forward to hearing from you.

Sincerely

Peng Yin, PhD

Professor of Epidemiology

National Center for Chronic and Noncommunicable Disease Control and Prevention

Chinese Center for Disease Control and Prevention

Round 2

Reviewer 1 Report

Comments have been addressed adequately